# Tabular Feature Discovery With Reasoning Type Exploration

## Abstract

Feature engineering for tabular data remains a critical yet challenging step in machine learning. Recently, large language models (LLMs) have been used to automatically generate new features by leveraging their vast knowledge. However, existing LLM-based approaches often produce overly simple or repetitive features, partly due to inherent biases in the transformations the LLM chooses and the lack of structured reasoning guidance during generation. In this paper, we propose a novel method REFEAT, which guides an LLM to discover diverse and informative features by leveraging multiple types of reasoning to steer the feature generation process. Experiments on 59 benchmark datasets demonstrate that our approach not only achieves higher predictive accuracy on average, but also discovers more diverse and meaningful features. These results highlight the promise of incorporating rich reasoning paradigms and adaptive strategy selection into LLM-driven feature discovery for tabular data.

## 1 Introduction

Automated feature engineering (AutoFE) has the potential to significantly improve model performance on tabular datasets by creating new predictive features, but it traditionally requires exploring a vast space of transformations Horn et al. (2019); Kanter & Veeramachaneni (2015); Khurana et al. (2016; 2018); Zhang et al. (2023). Recent advances in large language models (LLMs) offer a new approach to this challenge: using LLMs' embedded knowledge to propose candidate features in natural language or code Han et al. (2024); Hollmann et al. (2023); Nam et al. (2024). Early work in this direction showed that providing an LLM with context about the dataset and task can yield meaningful, human-interpretable features that boost prediction accuracy. For example, an LLM can be prompted with a dataset's description and asked to suggest a formula or grouping that might correlate with the target variable, producing features that a human expert might derive Hollmann et al. (2023).

However, existing LLM-based feature engineering methods have important limitations. One issue is that LLMs tend to generate simple or repetitive features Küken et al. (2024). Due to implicit biases in LLM training, they often overuse basic operations (e.g. linear combinations) and rarely utilize more complex transformations, which can lead to diminishing returns. Another limitation is the lack of structured reasoning guidance in current approaches. Most methods prompt the LLM in a relatively straightforward way - for instance, asking for a new feature given the task - and use validation performance to accept or reject features Abhyankar et al. (2025). They do not explicitly encourage the LLM to reason in diverse ways about the data. As a result, the generation process may miss creative insights; without guidance, the LLM might default to surface-level patterns or well-trod heuristics, rather than considering deeper relationships (causal factors, analogies to known problems, hypothetical what-if scenarios, etc.).

In this work, we propose a new method REFEAT (**R**easoning type **E**xploration for **Feat**ure discovery) to address these challenges by adopting multiple reasoning strategies with adaptive prompt selection for LLM-driven feature discovery. Our key insight is that different reasoning paradigms can inspire the LLM to generate different kinds of features, and that an adaptive approach can decide which type of reasoning is most fruitful for a given task. Specifically, we design reasoning-type-specific meta-prompts that each frame the feature generation task through a particular lens of reasoning: inductive, deductive, abductive, analogical, counterfactual, and causal reasoning Peirce (1903); Neuberg (2003); Gentner (1983); Lewis (1973). For example, an inductive prompt might say, "Examine these examples

and hypothesize a new feature that distinguishes the classes based on observed patterns," whereas a causal prompt might ask, "Identify a factor that could be a direct cause of the target, and formulate a feature to measure that cause." By crafting prompts in this way, we guide LLM to follow different reasoning paths, with the aim of generating a wider variety of candidate features - including those that a single, generic prompting approach might overlook.

Moreover, our approach includes an adaptive prompt selection mechanism that learns which reasoning strategy works best over time. We cast the choice of reasoning prompt as a multi-armed bandit problem Slivkins et al. (2019), where each "arm" corresponds to one of the reasoning types. At each iteration of feature generation, the bandit must decide which type of reasoning prompt to deploy, where the reward signal for the bandit comes from the performance of the new feature on a holdout validation set. This bandit-driven approach enables dynamic guidance of the LLM: unlike static prompting or fixed cycles of reasoning types, the strategy adapts based on which prompts are actually leading to good features for each task.

We conduct comprehensive experiments on 59 real-world tabular datasets from OpenML, spanning binary classification, multi-class classification, and regression tasks. Results show that our method consistently outperforms several baseline methods, including traditional automated feature engineering tools and LLM-based baselines. We also found that the proposed method discovers features that have higher complexity and greater mutual information with the target on average, indicating they carry more novel signal. Codes will be available soon via GitHub repository.

## 2 RELATED WORK

### 2.1 AUTOMATED FEATURE ENGINEERING

The challenge of automatically generating new features from raw tabular data has been studied in the AutoML community. Traditional AutoFE approaches do not use language models, but rather algorithmic search over transformation compositions Fan et al. (2010); Kanter & Veeramachaneni (2015); Khurana et al. (2016; 2018); Luo et al. (2019); Shi et al. (2020). For example, OpenFE is a recent tool that integrates a feature boosting method with a two-stage pruning strategy to efficiently evaluate candidate features Zhang et al. (2023). It incrementally builds and tests new features, aiming to achieve expert-level performance without human intervention. Another example is AutoFeat Horn et al. (2019), which generates a large pool of non-linear transformed features (combinations of original features through arithmetic, polynomial, or trigonometric functions) and then selects a subset based on model improvement.

Such methods can discover complex features, but they often require brute-force exploration of the search space and lack semantic understanding of the domain Hollmann et al. (2023). They treat feature construction as purely a mathematical optimization, which can miss intuition-driven features that humans might create using domain knowledge. Moreover, these methods might struggle when the space of possible transformations is huge, since they must handcraft or enumerate candidate operations Overman et al. (2024); Qi et al. (2023).

### 2.2 LLM-BASED FEATURE ENGINEERING

With the rise of powerful LLMs, researchers have started to leverage their knowledge and reasoning abilities for feature engineering in tabular data. One of the pioneering works is CAAFE Hollmann et al. (2023). CAAFE iteratively queries an LLM (such as GPT-3) to propose new features by providing it with the dataset's description, feature meanings, and the prediction task. This method showed that even a relatively simple prompting strategy can yield performance improvements on many datasets, demonstrating the LLM's ability to produce semantically meaningful features. Another example is OCTree Nam et al. (2024), which provides a mechanism for the LLM to get feedback from past experiments in a human-readable form. It translates the performance of previously generated features into a decision tree representation, then feeds that textual summary back into the LLM for the next round.

There are also notable efforts exploring feature engineering in few-shot and unsupervised learning contexts. FeatLLM Han et al. (2024), designed for few-shot learning scenarios, prompts an LLM with a handful of labeled examples and asks it to extract rules or conditions that differentiate the classes. More recently, TST-LLM Han et al. (2025) has been proposed for improving self-supervised

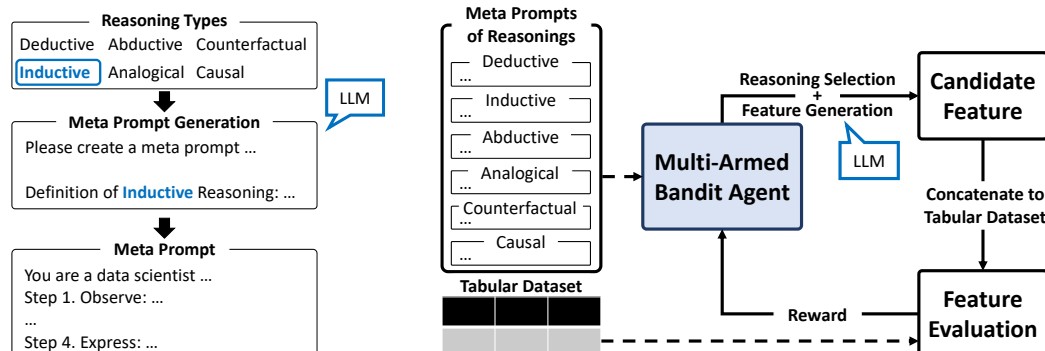

(a) Reasoning-Aware Prompt Generation     (b) Dynamic Prompt Selection with Multi-Armed Bandits

Figure 1: Illustration of REFEAT. (a) Meta prompts for each reasoning type are generated using an LLM. These prompts provide guidance on the reasoning steps associated with each strategy. (b) A multi-armed bandit agent selects a reasoning strategy and generates a feature using the LLM and the corresponding meta prompt. The generated feature is concatenated with the original tabular dataset, and the evaluation gain is fed back to the bandit agent as a reward.

learning on tabular data. TST-LLM similarly uses an LLM to generate additional features or labels that are relevant to a given task even before seeing the true labels.

In summary, the literature shows a progression from direct prompting of general LLMs to more sophisticated integration of LLMs in the feature generation loop. Our work builds on these ideas by using an LLM as the generator but focuses on a new dimension: guiding the mode of reasoning the LLM uses. By introducing multiple reasoning-oriented prompts, we aim to circumvent the mode collapse or bias problem and to inject a richer set of heuristic patterns for the model to draw on.

## 3 METHOD

**Problem Formulation.** Let $D = \{(x_i, y_i)\}_{i=1}^N$ be a dataset in which each raw instance $x_i \in \mathbb{R}^d$ is described by $d$ original features and the corresponding target $y_i$ is either categorical (classification) or real-valued (regression). A candidate transformation $g \in \mathcal{F}$ is a function that maps the original feature vector to a new scalar: $\varphi = g(x)$. Our aim is to construct a sequence of transformations $G = (g_1, \ldots, g_K)$ and augment the data with new features $X^+ = [X, \varphi_1, \ldots, \varphi_K]$. Denote by $\mathcal{M}$ a chosen downstream learner (e.g., linear regression or XGBoost Chen & Guestrin (2016)) and by $\mathcal{L}(\cdot)$ its empirical risk - accuracy loss for classification or squared error for regression - estimated on a validation split $D_{\text{val}}$. The main objective is to discover

$$G^\star = \arg\min_{G \subset \mathcal{F}} \mathcal{L}\big(\mathcal{M}; X^+(G), D_{\text{val}}\big). \qquad (1)$$

**Overview.** The proposed method REFEAT includes a reasoning-aware prompt library with an adaptive controller to navigate the transformation space efficiently. At each iteration, our method's controller (1) selects an appropriate reasoning type, (2) instantiates the corresponding meta-prompt using dataset context, (3) queries the LLM to produce an executable transformation, (4) evaluates the transformation's marginal utility on the validation set, and (5) updates its controller with the observed gain. Because different reasoning types favor different kinds of transformations, an adaptive scheduler is essential: it learns which reasoning type is most promising for the current task while still reserving budget to probe less-tried alternatives. Concretely, we model prompt selection as a multi-armed bandit whose arms correspond to reasoning types and whose rewards are validation-set performance gains. Figure 1 illustrates the overall workflow of REFEAT.

### 3.1 REASONING-AWARE PROMPT GENERATION

The first stage of our approach is constructing a list of meta-prompts that translate high-level reasoning principles into concrete instructions for the LLM. We guide the LLM with instructions corresponding

to six reasoning types drawn from cognitive science and logical problem solving Peirce (1903); Neuberg (2003); Gentner (1983); Lewis (1973). Below, we briefly define each reasoning type and how it influences feature generation:

- **Deductive Reasoning.** Derives new features by applying general rules or mathematical principles that are known to hold. The focus is on generating logically valid transformations that follow from established premises.

- **Inductive Reasoning.** Infers features by generalizing patterns observed in a few-shot examples. It emphasizes discovering trends or correlations that appear consistently across the data samples.

- **Abductive Reasoning.** Proposes features that represent the most plausible hidden causes explaining the observed data. It generates hypotheses that could account for surprising or non-obvious relationships in the dataset.

- **Analogical Reasoning.** Creates new features by drawing parallels to known constructs or transformations from similar domains. It transfers relational patterns or formulas from analogous situations to the current context.

- **Counterfactual Reasoning.** Constructs features by imagining alternative scenarios where certain variables take different values. It reflects how outcomes might change under hypothetical interventions or modifications.

- **Causal Reasoning.** Generates features that express potential cause-and-effect relationships among variables. It aims to capture mechanisms or mediators that explain how one variable influences another.

For each reasoning type, we pre-define a natural-language template which guides the LLM to favor the kind of logical operation characteristic of that reasoning type. To minimize researcher bias, we bootstrap these prompts with GPT-4.1-mini Achiam et al. (2023) itself; the model receives the formal definition of the reasoning mode and is asked to produce a short template that instructs the model to adopt that reasoning perspective and returns a transformation. During feature discovery, the template is filled with task-specific context: task descriptions, feature names, feature descriptions, few-shot examples, performance results from previous iteration's feature discovery, and any constraints on allowable libraries (see Appendix A for full prompt examples). By explicitly framing the generation step through a chosen reasoning lens, we encourage the LLM to traverse qualitatively distinct regions (e.g., inductive prompts tend to propose empirically driven aggregates, while causal prompts seek transformations suggestive of mechanistic influence).

### 3.2 DYNAMIC PROMPT SELECTION WITH MULTI-ARMED BANDITS

Next, we frame the selection of reasoning types as a multi-armed bandit problem Slivkins et al. (2019), where each "arm" corresponds to one of the six reasoning types. The goal is to adaptively allocate more trials to the reasoning modes that yield better features, while still exploring all options. Let $\mathcal{R} = \{ded, ind, abd, ana, cnt, cau\}$ denotes the set of reasoning categories and $Q_t(r)$ denote the estimated value of category $r \in \mathcal{R}$ after $t$ feature evaluations. We employ an $\varepsilon$-greedy bandit strategy Kuleshov & Precup (2014) with a decaying exploration rate:

$$r_t = \begin{cases} \arg\max_{r \in \mathcal{R}} Q_t(r), \text{with prob. } 1 - \varepsilon_t, \\ \text{a uniformly random } r \in \mathcal{R}, \text{with prob. } \varepsilon_t. \end{cases} \tag{2}$$

Specifically, at the beginning of the feature generation process ($t = 0$), the exploration probability $\varepsilon_t$ is set to 1 (i.e. 100% exploration, which means the reasoning type for the first iteration is chosen uniformly at random). As iterations proceed, $\varepsilon_t$ is linearly decayed from 1 to 0, gradually shifting from exploration to exploitation. Every category is given an optimistic value for the first time visit $Q_0(r_t)$ so that untried reasoning types are sampled early.

After choosing a reasoning type for the current iteration, we retrieve the corresponding meta-prompt constructed in Section 3.1 and query the LLM to generate candidate features $\varphi$. Each proposed feature is essentially a definition (e.g., a formula or transformation) that can be applied to the dataset's existing features. We immediately evaluate the utility of each generated feature $\varphi$ along with original features using a validation set. Specifically, we train a simple predictive model (e.g., linear regression or XGBoost) on the training set using the original features plus each new feature

(i.e., $[X, \varphi]$), and measure the performance on the validation set. We compare this to a baseline model trained on the original features alone. The performance gain $\Delta_t$ associated with feature $\varphi$ at iteration $t$ is computed as the difference in validation metrics (i.e., accuracy gain ratio for classification or RMSE reduction ratio for regression). This evaluation procedure treats each new feature individually, measuring its marginal benefit when added to the model.

Then, the controller updates its estimate for the current reasoning type via

$$Q_{t+1}(r_t) \leftarrow Q_t(r_t) + \alpha\big(\Delta_t - Q_t(r_t)\big), \tag{3}$$

where $\alpha$ is a learning rate; all other $Q$ values remain unchanged. Bounded reward magnitudes and the short bandit horizon makes this simple update rule sufficient Sutton et al. (1998); Gray et al. (2020). In practice we set $\alpha$ to the harmonic step $1/n_{r_t}$, where $n_{r_t}$ counts how many times category $r_t$ has been chosen.

Over the course of multiple iterations (e.g., 20), we maintain a global list of all generated features and their validation performance gains. After all iterations, we rank the candidate features by their $\Delta_t$ values and select the top-$K$ features overall. The final chosen features are then added to the original dataset, yielding an augmented feature matrix $X^+ = [X, \varphi_1, \ldots, \varphi_K]$. This augmented dataset is used to train the final predictive model whose performance is reported on the test set.

## 4 RESULTS

### 4.1 PERFORMANCE EVALUATION

**Datasets.** We evaluate on 59 publicly-available tabular datasets sourced from OpenML[1]. The corpus spans 51 classification tasks (i.e., binary and multi-class) and 8 regression tasks, covering domains such as finance (e.g., *credit-g* Kadra et al. (2021), *bank* Moro et al. (2014)), health (e.g., *diabetes* Smith et al. (1988)), scientific simulations (e.g., *climate-model-simulation-crashes* Lucas et al. (2013)), and sensor logs (e.g., *gesture* Madeo et al. (2013)). The full list is provided in Appendix B. Task metadata (i.e., input schema, feature descriptions, suggested target) is extracted from OpenML's official documentation.

**Baselines.** We compare REFEAT with GPT-4.1-mini against six baselines: the raw dataset with no additional features (i.e., ORIGINAL); conventional AutoFE systems AUTOFEAT Horn et al. (2019) and OPENFE Zhang et al. (2023); and three LLM-based methods—CAAFE Hollmann et al. (2023), FEATLLM Han et al. (2024), and OCTREE Nam et al. (2024). Each method is allowed to generate up to 10 features; we retain candidates that yield the highest validation improvement when concatenated to the original features.

**Evaluation.** For classification we report accuracy; for regression we report root-mean-squared error (RMSE, lower is better). We employ two downstream models of contrasting capacity: (1) logistic/linear regression to test linear separability, and (2) XGBoost to examine gains under a powerful non-linear learner. To facilitate dataset-wise aggregation, we compute two aggregate statistics: (1) the win matrix, where entry $W[i, j]$ is the ratio of datasets on which method $i$ strictly outperforms $j$ (ties excluded); and (2) the mean and median relative performance gain. In detail, we define the relative gain of method $i$ on each dataset as

$$\Delta_i^{(cls)} = \frac{\text{Acc}(i) - \text{Acc}(\text{orig})}{\text{Acc}(\text{orig})}, \qquad \Delta_i^{(reg)} = \frac{\text{RMSE}(\text{orig}) - \text{RMSE}(i)}{\text{RMSE}(\text{orig})}, \tag{4}$$

where $\Delta_i^{(cls)}$ is for classification tasks and $\Delta_i^{(reg)}$ is for regression tasks.

**Results.** Figure 2 shows the pair-wise win matrix averaged over the two learners, while Table 1 reports aggregate improvement statistics. Our model achieves the highest average win ratio and wins against every baseline in more than 67.04% of pair-wise comparisons. Concretely, across the 59 classification tasks it boosts performance by a mean of +4.65% and a median of +1.17%

---

[1]All datasets and baseline models used are publicly available for research and used in accordance with their licenses.

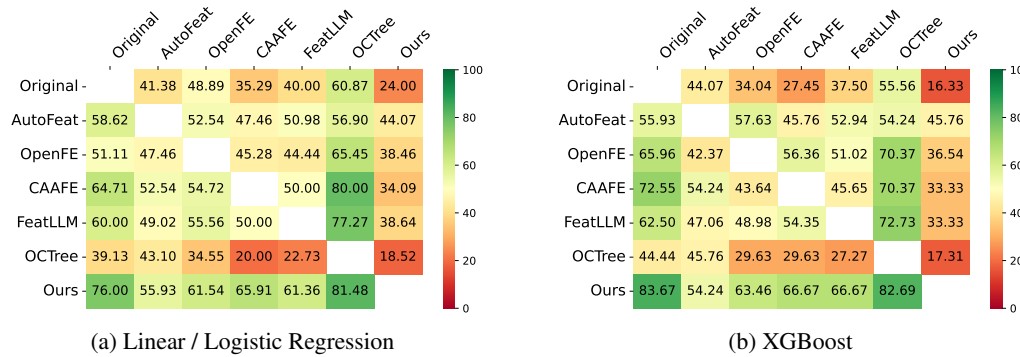

Figure 2: Win matrices comparing feature engineering methods against each other with (a) linear/logistic regression and (b) XGBoost. Tabular feature engineering methods are aligned on the x-axis and the y-axis while the numbers represent the winning ratio of the x-axis model against the y-axis model. Full results are reported in the Appendix H.

| Gain (%) | AutoFeat | OpenFE | CAAFE | FeatLLM | OCTree | Ours |
|---|---|---|---|---|---|---|
| Mean | -13.99 | -3.06 | 1.74 | 1.59 | -0.22 | **5.66** |
| Median | 0.89 | 0.03 | 0.37 | 0.00 | -0.11 | **1.23** |

(a) Linear / Logistic Regression

| Gain (%) | AutoFeat | OpenFE | CAAFE | FeatLLM | OCTree | Ours |
|---|---|---|---|---|---|---|
| Mean | -10.47 | -2.26 | 1.90 | 1.67 | -1.99 | **5.42** |
| Median | 0.40 | 0.00 | 0.23 | 0.00 | 0.00 | **0.85** |

(b) XGBoost

Table 1: Mean and median gain (%) of tabular feature engineering methods compared to ORIGINAL, using (a) linear/logistic regression and (b) XGboost. The best results are highlighted in bold.

relative to ORIGINAL. These gains persist for both linear and tree-based learners, underscoring that the discovered features complement a variety of model architectures. The out-performance over AUTOFEAT and OPENFE indicates that LLM guidance supplies richer semantic transformations than heuristic operator search, while the margin over recent LLM baselines highlights the value of reasoning diversity and adaptive prompt selection. Note that for AUTOFEAT and OPENFE, overfitting occurred on some small-sized datasets, leading to a significant drop in performance on average. Steering the LLM with multiple reasoning lenses and a simple bandit policy consistently uncovers features that generalize across heterogeneous datasets and model families.

## 4.2 ABLATION STUDY

We next dissect our design choices by comparing four reduced variants on the same 59 dataset benchmark:

- **Baseline (No-Guide)**: removes reasoning prompts and instead follows a generic feature-engineering prompt (functionally equivalent to CAAFE);
- **Single-Type**: runs six separate models, each fixed to one reasoning type (e.g., *Abductive*);
- **Uniform-Select**: cycles through all reasoning types with equal probability, foregoing bandit adaptation; and
- **Full**: Our model with full components.

All other hyper-parameters, feature budgets, and evaluation settings mirror the previous performance evaluation experiment.

**Results.** Table 2 summarizes the results. Our model delivers the largest mean (+5.42%) and median (+0.85%) performance gain . Removing reasoning guidance (No-Guide) cuts the average gain more than half, confirming that explicitly framing the task through distinct reasoning paradigms is crucial. Among single-type variants, Analogical and Causal prompts perform best but still trail the full model by 3.68%–4.85% in average gain, suggesting complementary benefits across types. Uniform-Select improves over any single fixed type yet lags the bandit strategy, indicating that adaptive exploitation of high-payoff reasoning modes is beneficial. In summary, we found that both components are necessary:

| Gain (%) | Baseline | Abductive | Analogical | Causal | Counterfactual | Deductive | Inductive | Uniform selection | Ours |
|---|---|---|---|---|---|---|---|---|---|
| Mean | 1.90 | 4.91 | 3.68 | 4.85 | 3.55 | 4.20 | 4.31 | 4.73 | **5.42** |
| Median | 0.23 | 0.53 | 0.61 | 0.74 | 0.27 | 0.53 | 0.38 | 0.53 | **0.85** |

Table 2: Mean and median performance gain of our full model and its ablations compared to ORIG-INAL. Baseline denotes feature engineering without a specific reasoning strategy, while abductive, analogical, causal, counterfactual, deductive, and inductive represent single-type reasoning, each corresponding to its respective reasoning type. Uniform selection applies all reasoning strategies with equal probability. The best results are highlighted in bold.

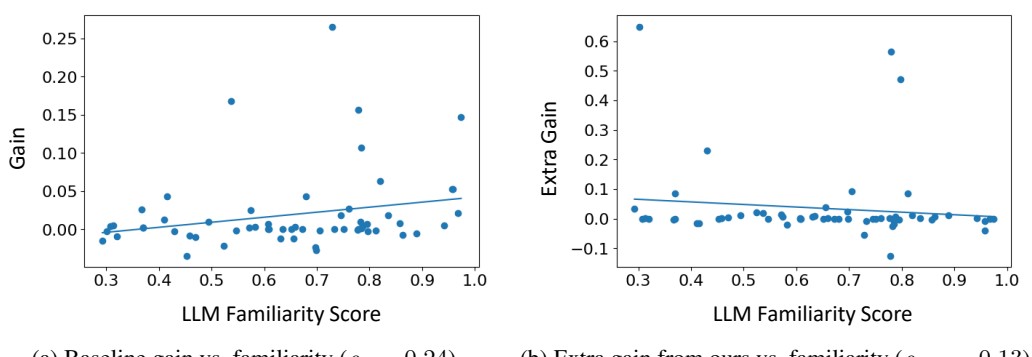

(a) Baseline gain vs. familiarity ($\rho_p = 0.24$)  (b) Extra gain from ours vs. familiarity ($\rho_p = -0.13$)

Figure 3: Scatter plots relating each dataset's LLM-familiarity score to the performance gain; (a) Baseline gain without any reasoning guidance, (b) Extra gain above the baseline achieved when our reasoning guidance is applied. Each dot corresponds to a single dataset and $\rho_p$ denotes the Pearson correlation.

reasoning-aware prompting diversifies the search space, and the bandit controller focuses exploration on the most promising reasoning patterns for a given dataset, together yielding robust performance improvements.

## 5 DISCUSSION

### 5.1 LLM FAMILIARITY VS. PERFORMANCE GAIN

A natural assumption is that if the LLM is more familiar with a dataset, it will generate better features. To examine this, we first analyze the relationship between LLM familiarity and the performance gain of a baseline LLM-based feature generation method without reasoning guidance over the original features (ORIGINAL). Following the strategy of prior work Bordt et al. (2024), we estimate how much the pretrained LLM "already knows'' about a tabular dataset via two completion tests:

- **Header completion.** We show the model the column headers and the first $t$ rows, then ask it to reproduce row $t+1$.
- **Row completion.** Starting from a random pivot row $r$, we prompt the LLM to generate row $r+1$ exactly as it appears in the file.

For each test, we compute the normalized Levenshtein distance between the model's output and the ground-truth row. Let $d_{\text{head},k}$ and $d_{\text{row},k}$ denote these distances for dataset $k$. We aggregate them into a single familiarity score

$$F_k = 1 - \frac{d_{\text{head},k} + d_{\text{row},k}}{2}, \tag{5}$$

where $F_k \in [0, 1]$ captures how closely the LLM's prior matches the dataset structure.

As shown in Figure 3a, there is a clear positive correlation between LLM familiarity and the performance gain of the baseline model, confirming that pretrained knowledge generally helps with

|                      | Num_ops | Depth |
| -------------------- | ------- | ----- |
| Baseline (No Guide)  | 3.74    | 2.43  |
| Ours                 | 4.87    | 3.11  |

Table 3: Structural properties of features generated by LLMs with and without reasoning guidance. Num_ops indicates the average number of operations per feature, while Depth indicates the depth of nested function calls

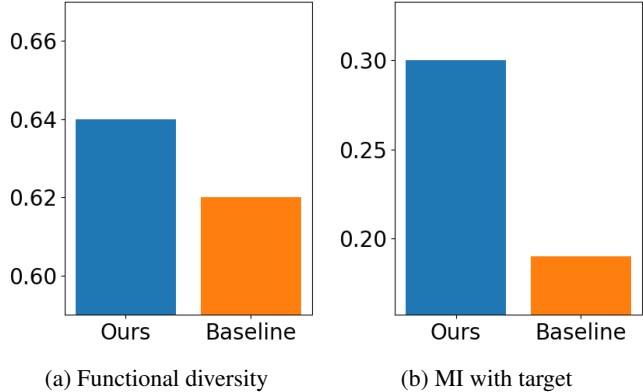

(a) Functional diversity          (b) MI with target

Figure 4: Semantic quality of features discovered with and without reasoning guidance.

feature generation. Then, The next question is: When does the guidance on reasoning strategy provide additional benefit over the baseline? To answer this, we compare LLM familiarity with the relative performance gain of our model over the baseline (Figure 3b). Interestingly, we observe a negative correlation - the relative gain is higher when the LLM is less familiar with the dataset. This indicates that our approach is particularly effective in domains where the LLM's pretrained knowledge is limited, as it can adaptively explore multiple reasoning strategies to compensate for the lack of prior knowledge.

## 5.2 COMPARISON OF GENERATED FEATURES

To understand how reasoning guidance influences the feature generation process, we compare the feature sets produced by our method and a baseline LLM-based approach without such guidance. Our analysis focuses on two dimensions: structural complexity and semantic quality of the features.

First, we examine the structural properties of the generated features, focusing on two metrics: the number of operations per feature (Num_ops) and the depth of nested function calls (Depth). As shown in Table 3, features generated by our method exhibit higher values for both metrics, reflecting a tendency to explore more complex and compositional transformations. This increased structural complexity may enable the model to capture patterns that are less accessible through simpler, shallow constructions, contributing to improved predictive performance.

Then, we assess the semantic quality of the features using two complementary measures: functional diversity and mutual information with target. Functional diversity is obtained by first computing the Pearson correlation for every pair of newly generated features, averaging those correlations, and then subtracting the result from one—so higher values indicate that the discovered features are less redundant and span a broader functional space. Mutual information with target is estimated by calculating the mutual information between each new feature and the ground-truth target label, then averaging these values across all discovered features to give an overall relevance score. Figure 4 shows that our method achieves higher scores on both metrics compared to the unguided baseline. These results suggest that reasoning-guided generation not only promotes richer functional exploration, but also yields features that are more informative with respect to the target task.

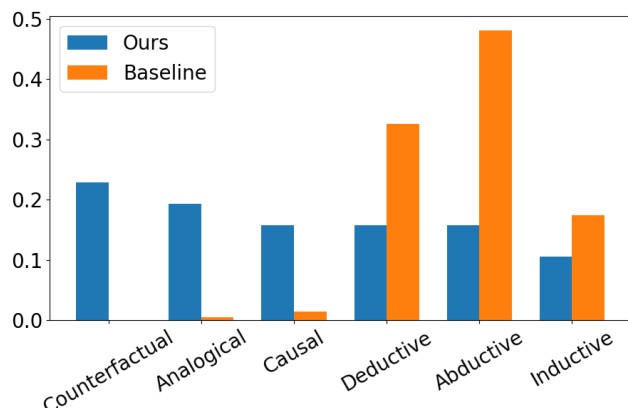

Figure 5: Distribution of reasoning types selected during feature discovery with and without reasoning guidance.

### 5.3 EFFECT OF DIFFERENT LLM BACKBONE

To test whether our reasoning-guided pipeline is tied to a specific LLM backbone model, we repeated the entire feature-generation procedure with two alternative backbones; DEEPSEEK-CHAT and LLAMA3.3-70B-INSTRUCT. Table 4 reports the relative performance increase with linear classifier over the original features. Note that results with XGBoost display the same trend (DeepSeek: +1.76% vs. +4.95%, Llama 3: –0.12% vs. +0.35%).

| Gain (%) | Baseline | Ours |
|---|---|---|
| DeepSeek-Chat | 1.97 (avg) / 0.69 (med) | **5.31 / 1.04** |
| Llama3.3-70B | 0.09 (avg) / 0.19 (med) | **4.32 / 0.78** |

Table 4: Average and median percentage improvement over the original feature set for different LLM backbones after feature discovery.

### 5.4 SELECTED REASONING TYPE BY DATASET

We analyze which reasoning strategies were finally selected across different datasets, as determined by our bandit. Figure 5 shows the distribution of selected strategies across datasets, comparing our model and the baseline. For the baseline, we extracted the chain-of-thought attached to each iteration, then fed that trace back to the LLM and asked it to label which of the six reasoning types it best matched. Compared to the baseline, which tends to heavily focus on a small subset of strategies (i.e., abductive, inductive, deductive), our method exhibits a more balanced usage across the full set of available reasoning styles. This suggests that our adaptive strategy selection mechanism effectively leverages different reasoning modes depending on the dataset characteristics.

## 6 CONCLUSION

This paper introduced REFEAT, a framework that discovers informative features for tabular learning. By casting feature construction as a reasoning-type allocation problem, our method dynamically selects reasoning types with a stochastic bandit, and finds candidate features that demonstrably boost validation performance. Comprehensive experiments on 59 OpenML datasets—far broader than those typically used in prior work—show that our method wins against contemporary baselines and achieves the largest mean and median performance gains for both linear and tree-based learners. These results confirm that reasoning diversity and data-driven prompt selection yield consistent improvements irrespective of the downstream model family, offering a practical path toward cost-effective, LLM-guided feature engineering.

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

# APPENDIX

## A    FULL PROMPT EXAMPLES

The below is the set of prompt examples for our proposed method, REFEAT, for each reasoning type. The blue-highlighted portion indicates the part of the prompt that varies depending on the reasoning type.

---

You are a data scientist tasked with discovering meaningful new features from tabular data. You will be shown a small number of data rows, each consisting of multiple columns. You are also given general domain knowledge or predefined rules. Your goal is to deductively derive useful new features by systematically applying known rules or definitions to the data. Given the examples and background knowledge, follow these steps:

Step 1. Recall: Identify relevant rules, formulas, or domain principles that could be applied to the given data.

Step 2. Apply: Use those rules to derive new features from existing columns.

Step 3. Explain: For each proposed feature, clearly explain the logic and rule that supports it.

Step 4. Express: Provide a formula or logic (in words or pseudo-code) for computing each new feature from the original columns.

Here below is the supported operation_type.

| Type | "Desc" | "Example" |
| — | — | — |
| "addition" | "A+B" | df["A"]+df["B"] |
| "subtraction" | "A-B" | df["A"]-df["B"] |
| "multiplication" | "AB" | df["A"]*df["B"] |
| "division" | "A/B" | df["A"]/df["B"] |
| "logarithm" | "log(A)" | np.log1p(df["A"]) |
| "nonlinear_transform" | "sqrt, square" | np.sqrt(df["A"]) |
| "flag_threshold" | "A>val" | (df["A"]>10).astype(int) |
| "combination_logical" | "A>val & B<val" | (df["A"]>5)&(df["B"]<3) |
| "groupby_agg" | "group-mean" | df.groupby("G")["A"].mean() |
| "interaction_logical" | "valflag" | df["A"]*(df["B"]>0) |
| "ranking" | "rank, qcut" | df["A"].rank() |

Task: <TASK DESCRIPTION>
Input columns: <FEATURE DESCRIPTION>

<FEW-SHOT EXAMPLES>

Previous features performance on validation:
[PREVIOUS FEATURE RESULTS]

Provide your reasoning step in <thinking> </thinking>, then return your new top-3 feature results (not in the original data) in the following JSON format:<Result> [{"feature_name": "...", "code": "..."}, ...] </Result> ONLY return a single Python EXPRESSION with pandas notation using 'df' for "code". Do NOT include "df['new']= ...", use only columns listed in the metadata, and do NOT include TARGET variable in the EXPRESSION.

---

Figure 6: Example prompt for feature discovery via deductive reasoning.

You are a data scientist tasked with discovering meaningful new features from tabular data.
You will be shown a small number of data rows, each consisting of multiple columns. Your goal is to inductively infer useful new features by identifying patterns and generalizing across the examples. Given the examples, follow these steps:

Step 1. Observe: Carefully examine how values across columns in the examples relate to each other across different rows. Explicitly refer to the corresponding examples for explanation.

Step 2. Induce: Propose new feature(s) that capture general patterns or derived quantities not explicitly represented in the original columns.

Step 3. Explain: For each proposed feature, clearly explain the reasoning behind it and how it generalizes across rows.

Step 4. Express: Provide a formula or logic (in words or pseudo-code) for computing each new feature from the original columns.

Here below is the supported operation_type.

| Type | "Desc" | "Example" |
| — | — | — |
| "addition" | "A+B" | df["A"]+df["B"] |
| "subtraction" | "A-B" | df["A"]-df["B"] |
| "multiplication" | "AB" | df["A"]*df["B"] |
| "division" | "A/B" | df["A"]/df["B"] |
| "logarithm" | "log(A)" | np.log1p(df["A"]) |
| "nonlinear_transform" | "sqrt, square" | np.sqrt(df["A"]) |
| "flag_threshold" | "A>val" | (df["A"]>10).astype(int) |
| "combination_logical" | "A>val & B<val" | (df["A"]>5)&(df["B"]<3) |
| "groupby_agg" | "group-mean" | df.groupby("G")["A"].mean() |
| "interaction_logical" | "valflag" | df["A"]*(df["B"]>0) |
| "ranking" | "rank, qcut" | df["A"].rank() |

Task: <TASK DESCRIPTION>
Input columns: <FEATURE DESCRIPTION>

<FEW-SHOT EXAMPLES>

Previous features performance on validation:
[PREVIOUS FEATURE RESULTS]

Provide your reasoning step in <thinking> </thinking>, then return your new top-3 feature results (not in the original data) in the following JSON format:<Result> [{"feature_name": "...", "code": "..."}, ...] </Result> ONLY return a single Python EXPRESSION with pandas notation using 'df' for "code". Do NOT include "df['new']= ...", use only columns listed in the metadata, and do NOT include TARGET variable in the EXPRESSION.

Figure 7: Example prompt for feature discovery via inductive reasoning.

You are a data scientist tasked with discovering meaningful new features from tabular data. You will be shown a small number of data rows, each consisting of multiple columns. Your goal is to use abductive reasoning to hypothesize the most plausible hidden causes or explanations for the observed data patterns, and propose new features accordingly. Given the examples, follow these steps:

Step 1. Observe: Carefully examine surprising or non-obvious patterns or correlations in the data.

Step 2. Hypothesize: Suggest possible latent variables or derived quantities that could plausibly explain the observed outcomes.

Step 3. Infer: Propose new features that would serve as those explanatory factors.

Step 4. Explain: Justify your hypothesis and explain how the new feature accounts for the observed data.

Step 5. Express: Provide a formula or logic (in words or pseudo-code) for computing the proposed feature from existing columns.

Here below is the supported operation_type.

| Type | "Desc" | "Example" |
| — | — | — |
| "addition" | "A+B" | df["A"]+df["B"] |
| "subtraction" | "A-B" | df["A"]-df["B"] |
| "multiplication" | "AB" | df["A"]*df["B"] |
| "division" | "A/B" | df["A"]/df["B"] |
| "logarithm" | "log(A)" | np.log1p(df["A"]) |
| "nonlinear_transform" | "sqrt, square" | np.sqrt(df["A"]) |
| "flag_threshold" | "A>val" | (df["A"]>10).astype(int) |
| "combination_logical" | "A>val & B<val" | (df["A"]>5)&(df["B"]<3) |
| "groupby_agg" | "group-mean" | df.groupby("G")["A"].mean() |
| "interaction_logical" | "valflag" | df["A"]*(df["B"]>0) |
| "ranking" | "rank, qcut" | df["A"].rank() |

Task: <TASK DESCRIPTION>
Input columns: <FEATURE DESCRIPTION>

<FEW-SHOT EXAMPLES>

Previous features performance on validation:
[PREVIOUS FEATURE RESULTS]

Provide your reasoning step in <thinking> </thinking>, then return your new top-3 feature results (not in the original data) in the following JSON format:<Result> [{"feature_name": "...", "code": "..."}, ...] </Result> ONLY return a single Python EXPRESSION with pandas notation using 'df' for "code". Do NOT include "df['new']= ...", use only columns listed in the metadata, and do NOT include TARGET variable in the EXPRESSION.

Figure 8: Example prompt for feature discovery via abductive reasoning.

You are a data scientist tasked with discovering meaningful new features from tabular data.
You will be shown a small number of data rows, each consisting of multiple columns. Your goal is to use analogical reasoning to propose new features by identifying relational patterns between data columns and applying similar transformations in new contexts. Given the examples, follow these steps:

Step 1. Identify: Find a relationship or transformation between two or more columns in one or more rows.

Step 2. Map: Check if similar relationships exist across other rows (i.e., establish analogies).

Step 3. Generalize: Propose a new feature that captures the underlying analogy consistently across rows.

Step 4. Explain: Describe the analogy and why the proposed feature fits the observed relational pattern.

Step 5. Express: Provide a formula or logic (in words or pseudo-code) for computing the new feature from the original columns.

Here below is the supported operation_type.

| Type | "Desc" | "Example" |
| — | — | — |
| "addition" | "A+B" | df["A"]+df["B"] |
| "subtraction" | "A-B" | df["A"]-df["B"] |
| "multiplication" | "AB" | df["A"]*df["B"] |
| "division" | "A/B" | df["A"]/df["B"] |
| "logarithm" | "log(A)" | np.log1p(df["A"]) |
| "nonlinear_transform" | "sqrt, square" | np.sqrt(df["A"]) |
| "flag_threshold" | "A>val" | (df["A"]>10).astype(int) |
| "combination_logical" | "A>val & B<val" | (df["A"]>5)&(df["B"]<3) |
| "groupby_agg" | "group-mean" | df.groupby("G")["A"].mean() |
| "interaction_logical" | "valflag" | df["A"]*(df["B"]>0) |
| "ranking" | "rank, qcut" | df["A"].rank() |

Task: <TASK DESCRIPTION>
Input columns: <FEATURE DESCRIPTION>

<FEW-SHOT EXAMPLES>

Previous features performance on validation:
[PREVIOUS FEATURE RESULTS]

Provide your reasoning step in <thinking> </thinking>, then return your new top-3 feature results (not in the original data) in the following JSON format:<Result> [{"feature_name": "...", "code": "..."}, ...] </Result> ONLY return a single Python EXPRESSION with pandas notation using 'df' for "code". Do NOT include "df['new']= ...", use only columns listed in the metadata, and do NOT include TARGET variable in the EXPRESSION.

Figure 9: Example prompt for feature discovery via analogical reasoning.

You are a data scientist tasked with discovering meaningful new features from tabular data.
You will be shown a small number of data rows, each consisting of multiple columns. Your goal is to use counterfactual reasoning to generate features that estimate what would have happened under alternative scenarios, by imagining changes to certain variables while keeping others fixed. Given the examples, follow these steps:

Step 1. Select: Choose a key variable that may be the cause of an observed outcome.

Step 2. Intervene: Imagine how the outcome would change if that variable had taken a different value (counterfactual intervention).

Step 3. Compare: Contrast the actual outcome with the counterfactual outcome.

Step 4. Construct: Propose a new feature that captures this difference or counterfactual scenario.

Step 5. Explain: Justify your reasoning and how this counterfactual feature could help model alternative behaviors or fairness.

Step 6. Express: Provide a formula or logic (in words or pseudo-code) for computing the counterfactual-based feature.

Here below is the supported operation_type.

| Type | "Desc" | "Example" |
| — | — | — |
| "addition" | "A+B" | df["A"]+df["B"] |
| "subtraction" | "A-B" | df["A"]-df["B"] |
| "multiplication" | "AB" | df["A"]*df["B"] |
| "division" | "A/B" | df["A"]/df["B"] |
| "logarithm" | "log(A)" | np.log1p(df["A"]) |
| "nonlinear_transform" | "sqrt, square" | np.sqrt(df["A"]) |
| "flag_threshold" | "A>val" | (df["A"]>10).astype(int) |
| "combination_logical" | "A>val & B<val" | (df["A"]>5)&(df["B"]<3) |
| "groupby_agg" | "group-mean" | df.groupby("G")["A"].mean() |
| "interaction_logical" | "valflag" | df["A"]*(df["B"]>0) |
| "ranking" | "rank, qcut" | df["A"].rank() |

Task: \<TASK DESCRIPTION>
Input columns: \<FEATURE DESCRIPTION>

\<FEW-SHOT EXAMPLES>

Previous features performance on validation:
[PREVIOUS FEATURE RESULTS]

Provide your reasoning step in \<thinking> \</thinking>, then return your new top-3 feature results (not in the original data) in the following JSON format:\<Result> [{"feature_name": "...", "code": "..."}, ...] \</Result> ONLY return a single Python EXPRESSION with pandas notation using 'df' for "code". Do NOT include "df['new']= ...", use only columns listed in the metadata, and do NOT include TARGET variable in the EXPRESSION.

Figure 10: Example prompt for feature discovery via counterfactual reasoning.

You are a data scientist tasked with discovering meaningful new features from tabular data.
You will be shown a small number of data rows, each consisting of multiple columns. Your goal is to use causal reasoning to infer and construct features that reflect causal relationships — that is, variables that directly influence others in the dataset. Given the examples, follow these steps:

Step 1. Identify: Look for potential causal relationships — i.e., where one variable might directly affect another.

Step 2. Justify: Provide reasoning or evidence from the data (e.g., monotonic patterns, interventions, domain knowledge) that supports the causal interpretation.

Step 3. Construct: Propose new features that represent either (a) the inferred cause, (b) the causal mechanism, or (c) a mediating factor between cause and effect.

Step 4. Explain: Clearly describe the assumed causal structure and how the new feature fits into it.

Step 5. Express: Provide a formula or logic (in words or pseudo-code) for computing the feature from original columns.

Here below is the supported operation_type.

| Type | "Desc" | "Example" |
| — | — | — |
| "addition" | "A+B" | df["A"]+df["B"] |
| "subtraction" | "A-B" | df["A"]-df["B"] |
| "multiplication" | "AB" | df["A"]*df["B"] |
| "division" | "A/B" | df["A"]/df["B"] |
| "logarithm" | "log(A)" | np.log1p(df["A"]) |
| "nonlinear_transform" | "sqrt, square" | np.sqrt(df["A"]) |
| "flag_threshold" | "A>val" | (df["A"]>10).astype(int) |
| "combination_logical" | "A>val & B<val" | (df["A"]>5)&(df["B"]<3) |
| "groupby_agg" | "group-mean" | df.groupby("G")["A"].mean() |
| "interaction_logical" | "valflag" | df["A"]*(df["B"]>0) |
| "ranking" | "rank, qcut" | df["A"].rank() |

Task: <TASK DESCRIPTION>
Input columns: <FEATURE DESCRIPTION>

<FEW-SHOT EXAMPLES>

Previous features performance on validation:
[PREVIOUS FEATURE RESULTS]

Provide your reasoning step in <thinking> </thinking>, then return your new top-3 feature results (not in the original data) in the following JSON format:<Result> [{"feature_name": "...", "code": "..."}, ...] </Result> ONLY return a single Python EXPRESSION with pandas notation using 'df' for "code". Do NOT include "df['new']= ...", use only columns listed in the metadata, and do NOT include TARGET variable in the EXPRESSION.

Figure 11: Example prompt for feature discovery via causal reasoning.

# B  DATASET DETAILS

The table below summarizes the basic statistics of the 59 benchmark datasets used in our work. It covers a total of 51 classification tasks and 8 regression tasks, with diverse ranges in both the number of samples and the number of features.

Table 5: Statistics of datasets used in our work.

| Data | Number of samples | Number of features (cat/num) | Task type |
|------|-------------------|------------------------------|-----------|
| adult | 48842 | 8/6 | binary classification |
| authorship | 841 | 0/69 | multi-class classification |
| balance-scale | 625 | 0/4 | multi-class classification |
| bank | 45211 | 9/7 | binary classification |
| blood | 748 | 0/4 | binary classification |
| breast-w | 683 | 0/9 | binary classification |
| car | 1728 | 6/0 | multi-class classification |
| churn | 5000 | 0/20 | binary classification |
| climate-model-simulation-crashes | 540 | 0/20 | binary classification |
| cmc | 1473 | 0/9 | multi-class classification |
| connect-4 | 67557 | 0/42 | multi-class classification |
| credit-a | 666 | 9/6 | binary classification |
| credit-g | 1000 | 13/7 | binary classification |
| cylinder-bands | 378 | 19/20 | binary classification |
| diabetes | 768 | 0/8 | binary classification |
| dmft | 661 | 2/2 | multi-class classification |
| dresses-sales | 500 | 11/1 | binary classification |
| electricity | 45312 | 0/8 | binary classification |
| eucalyptus | 641 | 5/14 | multi-class classification |
| first-order-theorem-proving | 6118 | 0/51 | multi-class classification |
| gesture | 9873 | 0/32 | multi-class classification |
| heart | 918 | 5/6 | binary classification |
| ilpd | 583 | 1/9 | binary classification |
| junglechess | 44819 | 0/6 | multi-class classification |
| kc2 | 522 | 0/21 | binary classification |
| kr-vs-kp | 3196 | 36/0 | binary classification |
| letter | 20000 | 0/16 | multi-class classification |
| mfeat-fourier | 2000 | 0/76 | multi-class classification |
| mfeat-karhunen | 2000 | 0/64 | multi-class classification |
| mfeat-morphological | 2000 | 0/6 | multi-class classification |
| mfeat-zernike | 2000 | 0/47 | multi-class classification |
| myocardial | 686 | 84/7 | binary classification |
| numerai | 96320 | 0/21 | binary classification |
| optdigits | 5620 | 0/64 | multi-class classification |
| ozone-level-8hr | 1847 | 0/72 | binary classification |
| pendigits | 10992 | 0/16 | multi-class classification |
| phishing | 11055 | 0/30 | binary classification |
| phoneme | 5404 | 0/5 | binary classification |
| qsar-biodeg | 1055 | 0/41 | binary classification |
| segment | 2310 | 0/19 | multi-class classification |
| spambase | 4601 | 0/57 | binary classification |
| splice | 3190 | 60/0 | multi-class classification |
| steel-plates-fault | 1941 | 0/27 | multi-class classification |
| texture | 5500 | 0/40 | multi-class classification |
| tic-tac-toe | 958 | 9/0 | binary classification |
| vehicle | 846 | 0/18 | multi-class classification |
| vowel | 990 | 2/10 | multi-class classification |
| wall-robot-navigation | 5456 | 0/24 | multi-class classification |
| wdbc | 569 | 0/30 | binary classification |
| wilt | 4839 | 0/5 | binary classification |
| wine | 6497 | 1/11 | multi-class classification |
| bike | 17379 | 0/12 | regression |
| crab | 3893 | 1/7 | regression |
| forest-fires | 517 | 2/10 | regression |
| grid-stability | 10000 | 0/12 | regression |
| housing | 20433 | 1/8 | regression |
| insurance | 1338 | 3/3 | regression |
| mpg | 398 | 1/6 | regression |
| satimage | 6430 | 0/36 | regression |

# C  IMPLEMENTATION DETAILS

Our proposed model, REFEAT, generates features over 20 iterations, using GPT-4.1-mini as the LLM backbone. From the generated candidates, we select the top-10 features that yield the highest validation performance for final evaluation. For the baselines, including AUTOFEAT, OPENFE, CAAFE, FEATLLM, and OCTREE, we followed the default settings provided in their respective

papers, except that we unify the LLM backbone and fix the number of generated features to 10 to ensure a fair comparison. The downstream predictors, including linear and logistic regression as well as XGBoost, were also used with their default parameters. Each dataset is split into training, validation, and test sets with a ratio of 6:2:2.

## D    DISTRIBUTION OF OPERATION TYPES

We compare the distribution of operation types used in the features generated by our method and the baseline LLM feature generation without reasoning guidance. Operation types include a mix of arithmetic operations (e.g., addition, multiplication), non-linear transformations (e.g., logarithm), logical constructs (e.g., threshold flags, logical combinations), and aggregations (e.g., group-by, ranking). As shown in Figure 12, the overall distribution patterns are broadly similar across both methods, without significant difference.

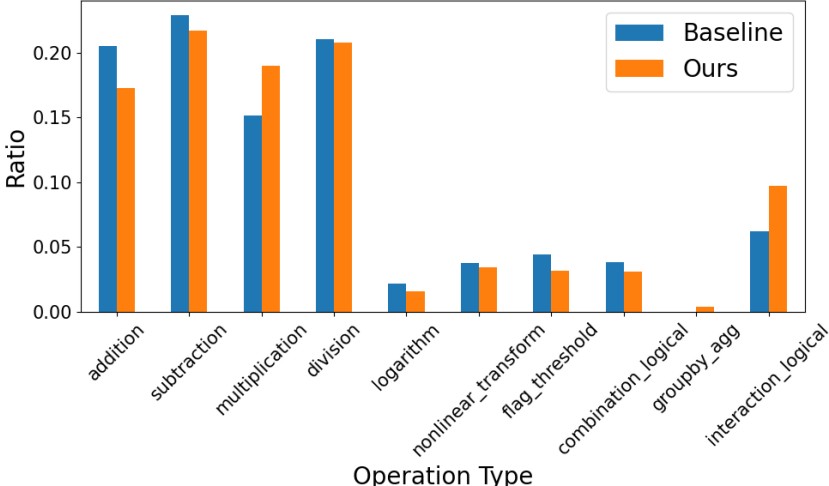

Figure 12: Histogram of operation type ratios used in feature discovery, comparing our model (REFEAT) and the baseline (without reasoning type exploration).

# E  QUALITATIVE ANALYSES

This section provides a comprehensive qualitative analysis of features generated by our REFEAT method compared to the baseline approach across 57 real-world datasets. Our analysis examines timing patterns and top-performing features. Key findings reveal that REFEAT achieves 2.95 times higher maximum performance gains over the validation set (47.84% vs 16.22%) and 2.41 times higher average gains (3.59% vs 1.49%) with similar feature generation volumes (i.e., # of selected features = 10) from the baseline.

## E.1  FEATURE GENERATION TIMING AND LEARNING PATTERNS

Analysis of iteration-based performance reveals REFEAT's superior learning dynamics throughout the feature discovery process. In early iterations (1-10) presented in Table 6, REFEAT already demonstrates substantial advantages with 3.26% average gain compared to the baseline's 1.42%. This gap widens in later iterations (11-20), where REFEAT achieves 3.94% compared to the baseline's 1.56%; unlike baseline approaches that show diminishing returns, REFEAT continues discovering better features throughout the exploration phase. Furthermore, the results demonstrate that our model converges faster by the inference strategy; our model's peak performance occurs on the iteration 7, where REFEAT reaches 7.41% gain; ours reached to the peak earlier than the baseline on the iteration 13, indicating effective bandit learning and adaptation.

| Phase | Iterations | REFEAT Gain | Relative Gain | Observation |
|-------|------------|-------------|---------------|-------------|
| Early Discovery | 1-10 | 3.26% (Avg.) | +128.9% | Quick adaptation |
| Late Optimization | 11-20 | 3.94% (Avg.) | +153.0% | Continued learning |
| Peak Performance (Baseline) | 13 | 2.51% | +17.9% | Baseline convergence |
| Peak Performance (REFEAT) | 7 | **7.41%** | **+437.9%** | Bandit convergence |

Table 6: Iteration-based performance analysis over the validation set. Bold is the best value.

## E.2  TOP PERFORMING FEATURES ANALYSIS

Table 7 presents the highest-performing features from both methods, revealing stark qualitative differences. REFEAT's top features demonstrate semantic meaningfulness, mathematical sophistication, and domain integration. The best-performing feature, responsiveness_index (47.84% gain), implements control theory principles through multi-ratio aggregation Schäfer et al. (2016). Features like high_bmi_smoker_flag (42.47% gain) capture compound risk factors with clinical relevance Taylor et al. (2019). In contrast, baseline features tend toward simple arithmetic operations with generic naming patterns. REFEAT features consistently exhibit interpretable, domain-relevant names that reflect understanding of field-specific relationships.

| Rank | Method | Feature Name | Gain (%) | Dataset | Reasoning Type |
|------|--------|--------------|----------|---------|----------------|
| 1 | REFEAT | responsiveness_index | 47.84 | grid-stability | Causal |
| 2 | REFEAT | high_bmi_smoker_flag | 42.47 | insurance | Counterfactual |
| 3 | REFEAT | weighted_tau_std | 40.83 | grid-stability | Causal |
| 1 | Baseline | DMFT_Trend | 16.22 | dmft | N/A |
| 2 | Baseline | Improvement_Male_Interaction | 10.81 | dmft | N/A |
| 3 | Baseline | Female_DMFT_Begin_Interaction | 10.81 | dmft | N/A |

Table 7: Top-5 performing features comparison over the validation set.

### E.3 KEY QUALITATIVE INSIGHTS

REFEAT demonstrates semantic intelligence by generating meaningful, interpretable feature names that reflect domain understanding. Features exhibit mathematical sophistication through complex multi-step operations rather than simple transformations. Domain awareness manifests in features that capture field-specific relationships and established principles. Also, different reasoning types demonstrate complementary strengths across domains, with the adaptive bandit strategy successfully identifying optimal reasoning for each dataset. The absence of a single dominant strategy validates the multi-reasoning approach and highlights the importance of reasoning diversity.

For practical implications, the generated features are production-ready with interpretable and auditable properties. The method successfully transfers across diverse application domains while producing expert-level quality features that typically require human domain expertise. Meanwhile, limitations exist too with the proposed method. The approach requires 20 iterations for convergence compared to single-shot methods, creating computational overhead. Quality depends on underlying LLM capabilities, and highly specialized domains may require additional expert guidance.

## F LIMITATION

The main limitation of our approach is the need for multiple interaction rounds with the LLM to identify the most suitable reasoning type for a given task. A promising remedy is to endow the LLM with a built-in "reasoning-type prior." In detail, one could fine-tune a LLM model on meta-examples where the input is a compact task description and the output is the reasoning strategy that ultimately produced the highest gain during past searches. We leave this fast-start variant and an investigation of how well it generalizes to entirely new domains as important future work.

Additionally, while our current work does not explicitly address biases in LLMs or training data, nor human-centered values, we acknowledge these risks as important considerations. A more thorough risk and ethics analysis, including human-centered evaluation and mitigation of unintended outcomes, remains an open direction for future work.

## G THE USE OF LLMS IN PAPER WRITING

We use LLMs in the paper writing only to polish writing.

## H  FULL RESULTS

| Dataset | Base | CAAFE | FeatLLM | OpenFE | OCTree | AutoFeat | Ours |
|---|---|---|---|---|---|---|---|
| Adult | 0.8543 | 0.8592 | 0.8611 | 0.8647 | 0.8533 | 0.7979 | 0.8614 |
| Authorship | 0.9882 | 0.9941 | 0.9941 | 0.9941 | 0.9941 | 1.0000 | 0.9941 |
| Balance-Scale | 0.8640 | 0.9920 | 1.0000 | 0.8640 | 0.8560 | 0.9333 | 1.0000 |
| Bank | 0.9013 | 0.9008 | 0.9038 | 0.8863 | 0.9015 | 0.9003 | 0.9029 |
| Blood | 0.7667 | 0.8000 | 0.8067 | 0.7867 | 0.7667 | 0.7656 | 0.8000 |
| Breast-W | 0.9635 | 0.9635 | 0.9635 | 0.9562 | 0.9562 | 0.9780 | 0.9635 |
| Car | 0.9017 | 0.9509 | 0.9566 | 0.0405 | 0.9046 | 0.9199 | 0.9364 |
| Churn | 0.8700 | 0.8720 | 0.8870 | 0.9000 | 0.8710 | 0.8637 | 0.8880 |
| Climate-Model-Simulation-Crashes | 0.9630 | 0.9537 | 0.9537 | 0.9630 | 0.9537 | 0.9383 | 0.9537 |
| Cmc | 0.5390 | 0.5864 | 0.5322 | 0.5627 | 0.5458 | 0.5617 | 0.5763 |
| Connect-4 | 0.6607 | 0.6953 | 0.6607 | 0.7259 | 0.6609 | 0.6613 | 0.6694 |
| Credit-A | 0.8134 | 0.8060 | 0.8134 | 0.8209 | 0.8134 | 0.8972 | 0.8284 |
| Credit-G | 0.7050 | 0.6950 | 0.7000 | 0.7200 | 0.7200 | 0.7867 | 0.7050 |
| Cylinder-Bands | 0.8026 | 0.6974 | 0.8158 | 0.8026 | 0.7368 | 0.9204 | 0.8026 |
| Diabetes | 0.7143 | 0.7208 | 0.7468 | 0.7273 | 0.7273 | 0.7652 | 0.7273 |
| Dmft | 0.2406 | 0.2782 | 0.2481 | 0.1429 | 0.2556 | 0.2390 | 0.2556 |
| Dresses-Sales | 0.6100 | 0.5900 | 0.5800 | 0.6000 | 0.6200 | 0.7833 | 0.6300 |
| Electricity | 0.7567 | 0.7573 | 0.7612 | 0.7597 | 0.7570 | 0.7527 | 0.7547 |
| Eucalyptus | 0.6357 | 0.6589 | 0.6357 | 0.6202 | 0.6124 | 0.5313 | 0.6589 |
| First-Order-Theorem-Proving | 0.4935 | 0.4935 | 0.5114 | 0.5033 | 0.4975 | 0.5210 | 0.4877 |
| Gesture | 0.4689 | 0.4759 | 0.4719 | 0.4800 | 0.4668 | 0.4266 | 0.4800 |
| Heart | 0.8859 | 0.8913 | 0.8859 | 0.8261 | 0.8804 | 0.8782 | 0.8804 |
| Ilpd | 0.6923 | 0.6923 | 0.6410 | 0.6923 | 0.7179 | 0.7278 | 0.7009 |
| Junglechess | 0.6756 | 0.7130 | 0.7282 | 0.6193 | 0.6778 | 0.7115 | 0.7063 |
| Kc2 | 0.8190 | 0.8000 | 0.8090 | 0.8190 | 0.8095 | 0.8526 | 0.8190 |
| Kr-Vs-Kp | 0.9625 | 0.9672 | 0.9625 | 0.7422 | 0.9641 | 0.9724 | 0.9688 |
| Letter | 0.7695 | 0.7893 | 0.8130 | 0.7903 | 0.7673 | 0.7108 | 0.7913 |
| Mfeat-Fourier | 0.8050 | 0.8075 | 0.7875 | 0.8100 | 0.7875 | 0.8417 | 0.8050 |
| Mfeat-Karhunen | 0.9575 | 0.9500 | 0.9350 | 0.9600 | 0.9596 | 1.0000 | 0.9550 |
| Mfeat-Morphological | 0.7200 | 0.7250 | 0.7200 | 0.7275 | 0.7050 | 0.6808 | 0.7250 |
| Mfeat-Zernike | 0.8325 | 0.8100 | 0.8225 | 0.8300 | 0.8125 | 0.7408 | 0.8450 |
| Myocardial | 0.7681 | 0.7899 | 0.7681 | 0.8043 | 0.7899 | 0.8832 | 0.7754 |
| Numerai | 0.5163 | 0.5166 | 0.5172 | 0.5165 | 0.5180 | 0.5247 | 0.5172 |
| Optdigits | 0.9689 | 0.9760 | 0.9671 | 0.9715 | 0.9689 | 0.9947 | 0.9795 |
| Ozone-Level-8Hr | 0.9378 | 0.9378 | 0.9378 | 0.9351 | 0.9297 | 0.9431 | 0.9405 |
| Pendigits | 0.9454 | 0.9673 | 0.9636 | 0.9686 | 0.9432 | 0.6644 | 0.9764 |
| Phishing | 0.9285 | 0.9313 | 0.9340 | 0.9249 | 0.9290 | 0.9430 | 0.9380 |
| Phoneme | 0.7364 | 0.7678 | 0.7835 | 0.7937 | 0.7401 | 0.8054 | 0.7567 |
| Qsar-Biodeg | 0.8626 | 0.8531 | 0.8578 | 0.8436 | 0.8531 | 0.9005 | 0.8626 |
| Segment | 0.9372 | 0.9589 | 0.9416 | 0.9610 | 0.9372 | 0.9286 | 0.9524 |
| Spambase | 0.9294 | 0.9294 | 0.9229 | 0.9338 | 0.9229 | 0.9457 | 0.9370 |
| Splice | 0.9451 | 0.9389 | 0.9436 | 0.5188 | 0.9326 | 0.9963 | 0.9404 |
| Steel-Plates-Fault | 0.7275 | 0.7481 | 0.7249 | 0.7326 | 0.7224 | 0.5687 | 0.7249 |
| Texture | 0.9927 | 0.9964 | 0.9927 | 0.9927 | 0.9927 | 0.9806 | 0.9927 |
| Tic-Tac-Toe | 0.9740 | 0.9427 | 0.9740 | 0.3490 | 0.9688 | 0.9826 | 0.9740 |
| Vehicle | 0.8176 | 0.8235 | 0.8471 | 0.8353 | 0.8294 | 0.7179 | 0.8176 |
| Vowel | 0.7727 | 0.9040 | 0.7626 | 0.7273 | 0.7525 | 0.9394 | 0.9091 |
| Wall-Robot-Navigation | 0.6923 | 0.8782 | 0.9139 | 0.9212 | 0.6969 | 0.9404 | 0.8480 |
| Wdbc | 0.9649 | 0.9737 | 0.9737 | 0.9649 | 0.9737 | 0.9677 | 0.9737 |
| Wilt | 0.9618 | 0.9855 | 0.9700 | 0.9866 | 0.9556 | 0.9459 | 0.9835 |
| Wine | 0.5415 | 0.5446 | 0.5492 | 0.5700 | 0.5392 | 0.5381 | 0.5538 |

Table 8: Evaluation results (Accuracy) of tabular feature engineering models using a linear model as the downstream predictor on 51 classification datasets.

| Dataset | Base | CAAFE | FeatLLM | OpenFE | OCTree | AutoFeat | Ours |
|---|---|---|---|---|---|---|---|
| Adult | 0.8538 | 0.8601 | 0.8512 | 0.8663 | 0.8536 | 0.8542 | 0.8611 |
| Authorship | 0.9882 | 0.9882 | 0.9882 | 0.9882 | 0.9882 | 1.0000 | 0.9882 |
| Balance-Scale | 0.8720 | 1.0000 | 1.0000 | 0.8720 | 0.8720 | 0.8933 | 1.0000 |
| Bank | 0.9014 | 0.9008 | 0.9021 | 0.8936 | 0.9018 | 0.8830 | 0.9028 |
| Blood | 0.7667 | 0.8000 | 0.7967 | 0.8000 | 0.7667 | 0.7656 | 0.8000 |
| Breast-W | 0.9635 | 0.9562 | 0.9635 | 0.9489 | 0.9562 | 0.9804 | 0.9635 |
| Car | 0.9364 | 0.9566 | 0.9682 | 0.0867 | 0.9306 | 0.9402 | 0.9566 |
| Churn | 0.8700 | 0.8720 | 0.8870 | 0.8960 | 0.8720 | 0.8643 | 0.8850 |
| Climate-Model-Simulation-Crashes | 0.9630 | 0.9537 | 0.9722 | 0.9630 | 0.9722 | 0.9846 | 0.9537 |
| Cmc | 0.5390 | 0.5966 | 0.5356 | 0.5661 | 0.5559 | 0.5493 | 0.5864 |
| Connect-4 | 0.6607 | 0.6955 | 0.6607 | 0.7258 | 0.6607 | 0.6613 | 0.6695 |
| Credit-A | 0.8134 | 0.7910 | 0.7836 | 0.8060 | 0.7910 | 0.9048 | 0.7910 |
| Credit-G | 0.7050 | 0.6900 | 0.7050 | 0.7000 | 0.7250 | 0.8033 | 0.7050 |
| Cylinder-Bands | 0.6974 | 0.6974 | 0.7895 | 0.7368 | 0.6974 | 1.0000 | 0.6974 |
| Diabetes | 0.7143 | 0.7273 | 0.7403 | 0.7273 | 0.7273 | 0.7848 | 0.7273 |
| Dmft | 0.2406 | 0.2782 | 0.2481 | 0.2030 | 0.2556 | 0.2369 | 0.2481 |
| Dresses-Sales | 0.5800 | 0.5600 | 0.5500 | 0.5500 | 0.6600 | 0.8367 | 0.5600 |
| Electricity | 0.7570 | 0.7575 | 0.7613 | 0.7568 | 0.7573 | 0.7570 | 0.7556 |
| Eucalyptus | 0.6124 | 0.6512 | 0.6124 | 0.6047 | 0.5969 | 0.8177 | 0.6589 |
| First-Order-Theorem-Proving | 0.4959 | 0.5008 | 0.5131 | 0.5000 | 0.4943 | 0.5264 | 0.4877 |
| Gesture | 0.4694 | 0.4739 | 0.4704 | 0.4820 | 0.4653 | 0.5306 | 0.4795 |
| Heart | 0.8859 | 0.8750 | 0.8804 | 0.4837 | 0.8750 | 0.8764 | 0.8804 |
| Ilpd | 0.6838 | 0.6752 | 0.6325 | 0.6923 | 0.6838 | 0.7163 | 0.7009 |
| Junglechess | 0.6758 | 0.7115 | 0.7281 | 0.6197 | 0.6780 | 0.7237 | 0.7063 |
| Kc2 | 0.8190 | 0.8000 | 0.8095 | 0.8190 | 0.7905 | 0.8558 | 0.8190 |
| Kr-Vs-Kp | 0.9797 | 0.9844 | 0.9797 | 0.8703 | 0.4781 | 0.9828 | 0.9859 |
| Letter | 0.7643 | 0.7855 | 0.8103 | 0.7838 | 0.7650 | 0.7538 | 0.7868 |
| Mfeat-Fourier | 0.7975 | 0.8075 | 0.7900 | 0.7975 | 0.7850 | 0.1000 | 0.7950 |
| Mfeat-Karhunen | 0.9450 | 0.9500 | 0.9350 | 0.9450 | 0.9317 | 1.0000 | 0.9525 |
| Mfeat-Morphological | 0.7325 | 0.7250 | 0.7300 | 0.7325 | 0.7200 | 0.7075 | 0.7275 |
| Mfeat-Zernike | 0.8250 | 0.8125 | 0.8225 | 0.8150 | 0.8075 | 0.6500 | 0.8400 |
| Myocardial | 0.7754 | 0.7754 | 0.7754 | 0.7754 | 0.7754 | 0.2214 | 0.7826 |
| Numerai | 0.5163 | 0.5184 | 0.5170 | 0.5170 | 0.5183 | 0.4948 | 0.5184 |
| Optdigits | 0.9689 | 0.9760 | 0.9582 | 0.9680 | 0.9609 | 0.0985 | 0.9724 |
| Ozone-Level-8Hr | 0.9432 | 0.9351 | 0.9432 | 0.9405 | 0.9432 | 0.9503 | 0.9378 |
| Pendigits | 0.9432 | 0.9668 | 0.9618 | 0.9650 | 0.9441 | 0.9503 | 0.9732 |
| Phishing | 0.9285 | 0.9322 | 0.9344 | 0.9236 | 0.9281 | 0.9426 | 0.9371 |
| Phoneme | 0.7364 | 0.7678 | 0.7789 | 0.7882 | 0.7401 | 0.8057 | 0.7558 |
| Qsar-Biodeg | 0.8531 | 0.8531 | 0.8578 | 0.8294 | 0.8578 | 0.8847 | 0.8531 |
| Segment | 0.9545 | 0.9567 | 0.9481 | 0.9654 | 0.9459 | 0.9452 | 0.9567 |
| Spambase | 0.9273 | 0.9349 | 0.9197 | 0.9327 | 0.9164 | 0.3942 | 0.9327 |
| Splice | 0.9201 | 0.9232 | 0.9185 | 0.6489 | 0.9216 | 0.9995 | 0.9248 |
| Steel-Plates-Fault | 0.7326 | 0.7352 | 0.7301 | 0.7275 | 0.7172 | 0.4794 | 0.7198 |
| Texture | 0.9964 | 0.9964 | 0.9964 | 0.9982 | 0.9964 | 0.9833 | 0.9964 |
| Tic-Tac-Toe | 0.9740 | 0.9688 | 0.9740 | 0.3490 | 0.6510 | 0.9861 | 0.9792 |
| Vehicle | 0.8235 | 0.8235 | 0.8412 | 0.8412 | 0.8294 | 0.6608 | 0.8235 |
| Vowel | 0.7828 | 0.9141 | 0.7980 | 0.7222 | 0.7828 | 0.9697 | 0.9293 |
| Wall-Robot-Navigation | 0.6932 | 0.8773 | 0.9112 | 0.9148 | 0.6859 | 0.8921 | 0.8397 |
| Wdbc | 0.9737 | 0.9737 | 0.9737 | 0.9737 | 0.6316 | 0.9795 | 0.9649 |
| Wilt | 0.9638 | 0.9886 | 0.9793 | 0.9866 | 0.9618 | 0.9480 | 0.9866 |
| Wine | 0.5362 | 0.5462 | 0.5408 | 0.5338 | 0.5362 | 0.5117 | 0.5477 |

Table 9: Evaluation results (Accuracy) of tabular feature engineering models using a XGboost model as the downstream predictor on 51 classification datasets.

| Dataset | Base | CAAFE | FeatLLM | OpenFE | OCTree | AutoFeat | Ours |
|---|---|---|---|---|---|---|---|
| Bike | 1.938E+04 | 1.935E+04 | N/A | 1.938E+04 | 1.940E+04 | 1.863E+04 | 8.438E+03 |
| Crab | 4.641 | 4.640 | N/A | 4.558 | 4.657 | 10.170 | 4.456 |
| Forest-Fires | 1.161E+04 | 1.174E+04 | N/A | 1.161E+04 | 1.155E+04 | 2.256E+03 | 1.164E+04 |
| Grid-Stability | 0.0005 | 0.0005 | N/A | 0.0005 | 0.0005 | 0.0013 | 0.0002 |
| Housing | 4.800E+09 | 4.800E+09 | N/A | 4.800E+09 | 4.805E+09 | 1.310E+10 | 4.400E+09 |
| Insurance | 3.360E+07 | 3.463E+07 | N/A | 3.360E+07 | 3.375E+07 | 1.200E+08 | 1.798E+07 |
| Mpg | 10.53 | 10.41 | N/A | 10.53 | 11.08 | 12.70 | 9.33 |
| Satimage | 1.387 | 1.387 | N/A | 1.135 | 1.395 | 4.895 | 1.065 |

Table 10: Evaluation results (RMSE) of tabular feature engineering models using a linear model as the downstream predictor on 8 regression datasets. Note that FeatLLM is applicable only to classification tasks.

| Dataset | Base | CAAFE | FeatLLM | OpenFE | OCTree | AutoFeat | Ours |
|---|---|---|---|---|---|---|---|
| Bike | 1.939E+04 | 1.939E+04 | N/A | 1.939E+04 | 1.940E+04 | 1.865E+04 | 8.442E+03 |
| Crab | 4.874 | 4.862 | N/A | 4.568 | 4.874 | 10.849 | 4.452 |
| Forest-Fires | 1.163E+04 | 1.165E+04 | N/A | 1.163E+04 | 1.157E+04 | 2.257E+03 | 1.166E+04 |
| Grid-Stability | 0.0005 | 0.0005 | N/A | 0.0003 | 0.0005 | 0.0000 | 0.0002 |
| Housing | 4.800E+09 | 4.810E+09 | N/A | 4.800E+09 | 4.811E+09 | 1.310E+10 | 4.400E+09 |
| Insurance | 3.360E+07 | 3.368E+07 | N/A | 3.360E+07 | 3.375E+07 | 1.200E+08 | 1.788E+07 |
| Mpg | 10.30 | 10.31 | N/A | 10.30 | 10.69 | 13.58 | 9.36 |
| Satimage | 1.387 | 1.390 | N/A | 1.163 | 1.387 | 1.387 | 1.069 |

Table 11: Evaluation results (RMSE) of tabular feature engineering models using a XGBoost model as the downstream predictor on 8 regression datasets. Note that FeatLLM is applicable only to classification tasks.

