# OpenReview forum: "Tabular Feature Discovery With Reasoning Type Exploration"
_ICLR.cc/2026/Conference — ICLR 2026 Conference Withdrawn Submission_

### Official Review · Reviewer_LYAw · 2025-10-30

**Soundness:** 1
**Presentation:** 4
**Contribution:** 3
**Rating:** 4
**Confidence:** 5

**Summary:**

The paper proposes an LLM-based automated feature engineering method, REFeat. REFeat prompts an LLM to generate new features, whereby various different prompts are used in each iteration to generate a feature. The various prompts are based on so-called reasoning types. The idea is that these different reasoning types induce different features and thus improve the feature generation process.  The various reasoning types are motivated by different ways of thinking about the problem, together with causal and logical literature. REFeat determines which prompt to use by starting with a uniform prior over the selection, which is adapted by a bandit over time. The authors evaluate REFeat on a collection of OpenML tasks and compare it to other AutoFE methods. The results show that it performs the best. Further ablations demonstrate the importance of prompt selection and the impact of data contamination.
The paper contributes a novel idea to LLM-based automated feature engineering that introduces reasoning types and prompt selection into the process. Moreover, it shows experiments to validate the idea and its individual novel components.

**Strengths:**

The idea proposed in the work seems very interesting, and combining prompt types and prompt selection introduces a novel optimization concept into LLM-based automated feature engineering. In general, it appears to be a significant methodological change compared to prior work. The motivation for the idea is valid, and bandit-based approaches can be promising. The paper is well-structured and easy to understand, thanks to its clear writing. The related work covers many of the relevant papers.

**Weaknesses:**

The core weakness of the paper is its experimental design, which I highly doubt will generalize to real-world applications of the compared methods. There are several reasons (detailed below) for this, most of which are known in the general literature on tabular data. Sadly, the literature on automated feature engineering often ignores this. As a result, **I want to highlight that the quality of the experimental design is in line with prior work on automated feature engineering**, which other reviewers found sufficient for publication.

Most of the flaws I perceive in the experimental design are also recognized by the community, as seen in TabArena [1] and its extended discussion of these flaws, as well as its overview of related work.

### Problems in Experimental Design
* 51 out of 59 datasets used in the evaluation are classification datasets, of which all are evaluated with accuracy. It is well known that accuracy as a metric is not a good choice for benchmarks (see [2]), especially for binary classification tasks without threshold tuning. Benchmarks commonly use ROC AUC for binary and log loss (or other proper scoring rules) for multiclass classification.
* The models used to evaluate the performance of the output of an AutoFE method are not tuned. This clearly misrepresents the actual benefit of the AutoFE tool, as it is unclear how much of the benefit from feature engineering would vanish if the models had been tuned. Prior work commonly restricted itself to an untuned LightGBM model for evaluation, so using two models is already a marginal improvement.
* Other AutoFE methods were used with default parameters as described in the appendix. This is problematic because some of these methods optimize for a non-accuracy-based target metric, while others optimize for other target models (LightGBM for OpenFE or TabPFN for CAAFE). In other words, the comparison with other models is potentially compromised by failing to ensure that the methods share the same optimization objectives. It becomes unfair in particular as REFeat directly optimizes for the target algorithm within its agentic loop.
* The work uses a 60:20:20 holdout split for each dataset. First, this makes the models and the AutoFE method extremely susceptible to overfitting, as holdout validation does not provide a reliable signal for AutoFE to generalize to test data (such as for tuning of models, see [1]). Second, the experiment has not been repeated for each dataset. Thus, the results are heavily impacted by noise from the dataset splits.
* Knowing that the experiment might be impacted by overfitting, the choice described in the paragraph from Line 229 also might tell us why REFeat can gain much compared to the traditional AutoFE baselines. That is, since REFeat selects the top-k features after evaluating 20 features, it cannot overfit to an incumbent state of the dataset during the optimization process (akin to overtuning [3]) while other baselines might do this. Moreover, picking the top-k best features actually does not guarantee that the final dataset has an improvement over the original dataset. The method never validates the final dataset, so there might be cases where it finds good features, but their combination is worse than the original dataset. This again is advantageous if the validation signal is unreliable due to holdout validation.
* There is no information on how the authors picked the benchmark datasets. Dataset collections from prior benchmarking efforts might be better suited. Moreover, many of the selected datasets might be unsuited for evaluating context-aware tabular feature engineering. For example, the collection includes image datasets (such as mfeat), so it seems unlikely that meaningful semantic features could be generated from pixel-based features.
* As far as I can tell, the results in Appendix D indicate that REFeat does not make the LLM generate more complex features (in contrast to the motivation and conclusion stated in the introduction). Moreover, it is questionable if the experiments would be more representative if LLMs could even improve over the baseline (akin to the results in Küken et al.).
* The difference in scale in Figure 3 makes the comparison of correlation problematic. Moreover, the plot shows that the outliers drive the correlation analysis, especially when Pearson correlation is used instead of Spearman.  Spearman is more robust to outliers. Furthermore, it would have been much more insightful to compare the performance gain over non-LLM-based AutoFE methods, or only report results on datasets below some threshold (akin to prior work from Bordt et al. and Küken et al.); otherwise, it is still unclear to what extent data contamination affects the results.
* Appendix C describes how many features were selected for different methods. Depending on the method, it is not a fair comparison to have REFeat create 20 and another method only 10, as this might represent different budgets. If one method only generates 10 (via an LLM) while REFeat creates 20 and takes the 10 best, then REFeat got twice the budget for evaluation. This is further complicated by comparing to baselines that work entirely differently internally. In other words, the experiments also need to control that a similar budget (in terms of resources or time) is used for all methods. Right now, this does not seem to be the case.


### Minor Feedback:
* I think the text would benefit from using \citep instead of \cite for references.
* I got the impression that almost all relevant details of the experimental design are in the appendix, even though it includes crucial aspects such as the splits and repetitions used in the experiments.




---

### References:
* [1] TabArena: A Living Benchmark for Machine Learning on Tabular Data, Erickson et al. 2025, https://neurips.cc/virtual/2025/poster/121499
* [2] The Case Against Accuracy Estimation for Comparing Induction Algorithms, Provost et al. 1998
* [3] Overtuning in Hyperparameter Optimization, Schneider et al. 2025

**Questions:**

* The code is not public; thus, it is essential to ask: Is the proposed method more usable than other AutoFE methods? (akin to the discussion in https://arxiv.org/abs/2508.13932)
* Is REFeat doing transductive or inductive learning? Depending on the answer, the compared methods are not quite on the same footing. AutoFE and OpenFE do inductive learning. OpenFE does transductive learning. From https://arxiv.org/abs/2508.13932: With inductive learning, we refer to a method that does not look at the test samples during training, as is the standard assumption for machine learning on tabular data. Transductive learning refers to a method that incorporates information from the test samples (without the labels) to influence feature engineering
* How was randomness controlled across experiments and in ablations?
* How does it occur that the scores for CAAFE (Table 1) and REFeat-Baseline (Table 2) are entirely identical? It makes sense that they should be similar but not identical, given noise and methodological differences. Is the same code used, the same data reported? The original version of CAAFE uses TabPFN and not an XGBoost model. Such a change might cause performance differences and misrepresent CAAFE's performance.
* From Figure 5, it appears that the bandit still produces a quite uniform selection distribution. What would this look like if one used many more iterations?

---

### Official Review · Reviewer_p7kc · 2025-10-30

**Soundness:** 2
**Presentation:** 2
**Contribution:** 2
**Rating:** 4
**Confidence:** 4

**Summary:**

This paper studies the effect of reasoning strategies on LLM-based automated feature engineering. It introduces REFEAT, an LLM-based AutoFE framework that dynamically selects the reasoning strategy using a multi-armed bandit algorithm and modifies the prompt instructions. The authors demonstrate through experiments on benchmark datasets that REFEAT achieves higher predictive accuracy on average for linear models and XGBoost, and it discovers more diverse and meaningful features.

**Strengths:**

It is a meaningful task to study the impact of prompt design on the performance of LLM-based AutoFE, and agentic systems in general, which may offer insights on the best practice when designing such systems. The methodology part of the paper is presented clearly. Adaptively selecting reasoning strategies benefits the generation of diverse features that may suit different datasets. I appreciate the efforts the authors have made on conducting experiments across 59 OpenML datasets and evaluating different LLM backbones.

**Weaknesses:**

While the motivation and methodology are clear, the technical novelty of this paper is somewhat limited. Automated prompt engineering approaches such as [1] may give more flexibility and even better performance. As a core of the work, the bandit design needs further exploration. Will it benefit from a different decay schedule of the exploration probability or a different algorithm like UCB [2]? It would be interesting to see further studies on this.

For the experiments, the number of augmented features has been set to a fixed number K per line 230, and it is unclear how this has been determined. A fixed number of features may not be optimal for all datasets. Moreover, the number of iterations 20 seems arbitrary, which seems a bit small to allow for sufficient exploration of the feature space and different reasoning strategies. A further analysis of these parameters is necessary.

Some key experimental details are missing. It is not reported how many repeated runs these results have been based upon. The standard errors of the results are not provided either. It is not stated how the parameters of downstream models have been selected, which could have a major impact on the performance. It would be great to also include a study on the LLM cost and computation cost of the framework.

There lacks explanation regarding the selection criteria for datasets. I do not think some of the datasets used in the experiments have meaningful text descriptions of features, e.g., kr-vs-kp, junglechess, and numerai. It is unclear how feature descriptions have been generated for these datasets. Mixing the results using datasets without semantic information of features would weaken the validity of claim on the effect of reasoning strategies. Besides, the study presented in Section 5.1 seems to suggest that the performance has been impacted by the leakage of dataset information in LLM pre-training.

The study in Section 5.4 is uninformative. It merely shows that strategy selection using a bandit is more uniform than baseline, while randomly selecting the strategies can attain similar results. It does not, however, show how the optimal reasoning strategy by the bandit varies across datasets. The explanations in Section E.2 are a bit vague and can be accompanied by specific feature examples.

In terms of paper presentation, the main text leaves quite some white spaces, e.g., around Table 3 and Figure 4. More compact formatting may help. Some citations could be changed to the parenthetical format.

[1] Zhou, Yongchao, et al. "Large language models are human-level prompt engineers." The eleventh international conference on learning representations. 2022.

[2] Auer, Peter, Nicolo Cesa-Bianchi, and Paul Fischer. "Finite-time analysis of the multiarmed bandit problem." Machine learning 47.2 (2002): 235-256.

**Questions:**

Please find weaknesses above.

Also, line 250 states that each method generates up to 10 features. Why is this inconsistent with the iteration limit 20 for REFEAT?

---

### Official Review · Reviewer_M8xr · 2025-10-31

**Soundness:** 3
**Presentation:** 3
**Contribution:** 3
**Rating:** 6
**Confidence:** 2

**Summary:**

This paper addresses the critical challenges in feature engineering for tabular data, specifically targeting the limitations of existing Large Language Model (LLM)-based approaches—such as the generation of overly simple or repetitive features and the lack of structured reasoning guidance. To tackle these issues, the authors propose REFEAT (Reasoning type Exploration for Feature discovery), a novel framework that integrates two core innovations: multi-type reasoning guidance and adaptive prompt selection.REFEAT designs meta-prompts for six reasoning paradigms derived from cognitive science and logical problem-solving: deductive, inductive, abductive, analogical, counterfactual, and causal reasoning. These prompts guide the LLM to generate features from distinct logical perspectives. Second, the framework models the selection of reasoning types as a multi-armed bandit problem, using an ε-greedy strategy with a decaying exploration rate. This allows dynamic adaptation to the most effective reasoning type for a given task, where the reward signal is derived from the performance gain of generated features on a holdout validation set.

The authors evaluate REFEAT on 59 real-world tabular datasets from OpenML, covering 51 classification tasks (binary and multi-class) and 8 regression tasks across domains like finance, healthcare, and scientific simulation. Experiments compare REFEAT against 6 baselines, including raw data (ORIGINAL), traditional automated feature engineering (AutoFE) tools (AutoFeat, OpenFE), and LLM-based methods (CAAFE, FeatLLM, OCTree). Results show that REFEAT consistently outperforms all baselines, achieving a win rate of over 67.04% in pairwise comparisons, with an average performance gain of +4.65% and a median gain of +1.17% across classification tasks.

**Strengths:**

(1) Innovative and Theoretically Grounded Design:The paper is the first to systematically incorporate six classical cognitive reasoning types into LLM-driven tabular feature engineering. This breaks the limitation of "generic prompting" in existing methods, providing structured logical guidance for feature generation and enriching the theoretical application of LLMs in structured data tasks.By framing reasoning type selection as a multi-armed bandit problem, the framework dynamically balances exploration (testing less-tried reasoning types) and exploitation (prioritizing effective types), avoiding the inefficiency of static prompting or fixed cycles. This data-driven optimization enhances the targeted nature of feature generation.
(2) Rigorous and Comprehensive Experimental Design:The use of 59 cross-domain datasets ensures the generalizability and reliability of results, addressing the limitation of narrow dataset scopes in prior work.The paper compares against 6 representative baselines (covering traditional AutoFE and state-of-the-art LLM methods) and uses a dual evaluation system—"win matrix" (to measure relative superiority) and "relative performance gain" (to quantify absolute improvements)—across both linear (logistic/linear regression) and non-linear (XGBoost) downstream models.Ablations on variants (No-Guide, Single-Type, Uniform-Select) clearly isolate the contributions of "reasoning guidance" and "adaptive selection," verifying that both components are indispensable for performance gains.
(3) In-Depth Result Analysis and Transparency:The paper analyzes feature quality (structural complexity, functional diversity, mutual information with the target) and LLM familiarity adaptation (showing stronger advantages in low-familiarity domains), revealing the intrinsic mechanisms behind REFEAT’s success and providing insights for future research.

**Weaknesses:**

(1) Computational Overhead and Efficiency Trade-Offs:REFEAT requires 20 iterations of LLM calls and model evaluations to generate features, resulting in higher computational costs compared to one-shot baseline methods (e.g., CAAFE). The paper does not discuss how to balance efficiency and performance, limiting its applicability to small-sample or real-time scenarios.Conduct a performance-efficiency curve analysis: Test iterations of 5, 10, 15, and 20 to identify the optimal iteration threshold and clarify how it correlates with dataset characteristics .Evaluate lightweight LLM backbones to reduce inference costs while ensuring acceptable accuracy loss.
(2) Lack of Discussion on LLM Bias and Ethical Risks: The paper acknowledges that LLM/training data biases and human-centered values are potential risks but provides no mitigation strategies. For tabular data in sensitive domains (e.g., credit scoring, medical diagnosis), biases (e.g., gender/race-related) could undermine the fairness and practicality of REFEAT.Suggest adding a fairness assessment.
(3) Insufficient Exploration of "Domain-Reasoning Type Adaptation":While the paper notes that REFEAT selects reasoning types based on datasets, it does not analyze which reasoning types are more suitable for specific domains/tasks (e.g., causal reasoning for healthcare data, inductive reasoning for financial data). This missed opportunity limits the ability to derive generalizable "domain-reasoning" rules.Suggestions adding cross-domain reasoning type distribution analysis.

**Questions:**

（1）The number of LLM calls per iteration of the paper, such as how many candidates each reasoning type generates and whether any post-processing is applied to the candidates; The random seed for the Bandit and the entire workflow, and whether multiple repetitions are performed for different random partitions/seeds with reporting of mean and variance.
（2）The paper defines the utility of each new feature as “the marginal gain when adding that single feature to the original set,” and updates the bandit and final top-K selection based on this metric. Please clarify and supplement the experiments. For instance, when using only “single-feature marginal gain” as the reward signal, could combinations with small individual gains but strong complementarity with existing new features be overlooked? Can comparisons be provided between the current method and subsequent candidates evaluated after each selected feature is added to the training set?

---

### Official Review · Reviewer_aDyL · 2025-11-04

**Soundness:** 3
**Presentation:** 2
**Contribution:** 2
**Rating:** 2
**Confidence:** 5

**Summary:**

This paper introduces an automated feature engineering method based on LLMs. The authors use reasoning to guide the generation of diverse features. The experiments on several datasets show improvement in task performance with selected model.

**Strengths:**

The paper is written with running examples. The core technical part is clear and easy to follow.

The problem of automated feature engineering is an important problem in data science. The idea of leveraging LLM to generate features based on dataset information is reasonable, and performing the appropriate data engineering operations can speed up the cycle.

**Weaknesses:**

For method design, the novelty is limited. Using reasoning (especially with prompting only) for feature generation is not a new topic. This work seems to be more of a combination of existing reasoning prompting paradigm with a controlling mechanism, not a fundamentally new mechanism for reasoning itself. The generation process, which heavily depends on the LLM's internal generation and reasoning variability, would be hard to reproduce and explain. Further, the method lacks any optimization or self-improvement strategy, it passively samples reasoning modes without deeper refinement of quality or stability.

For presentation, the authors may refer to CAFFE and similar works to refine their problem definition and symbols.

For experiments, LLMs’ understanding and reasoning abilities on tabular data remain inconsistent and model-dependent. That means using the GPT api as the only model to prove the effectiveness of this method is far from enough. More models like qwen or mistral should be included. The ablation study is not complete. And the cost should be discussed as a common concern for LLM-based feature generation methods.

**Questions:**

See above.

---

### Note · Authors · 2025-11-23

**Comment:**

Thank you for the review. We will reflect the review content and improve the paper.

**Withdrawal Confirmation:**

I have read and agree with the venue's withdrawal policy on behalf of myself and my co-authors.